# Direct Evaluation of Mixed Mode I+II Cohesive Laws of Wood by Coupling MMB Test with DIC

**DOI:** 10.3390/ma14020374

**Published:** 2021-01-14

**Authors:** Jorge Oliveira, José Xavier, Fábio Pereira, José Morais, Marcelo de Moura

**Affiliations:** 1CITAB, School of Science and Technology, University of Trás-os-Montes e Alto Douro, 5001-801 Vila Real, Portugal; jqomarcelo@demad.estv.ipv.pt (J.O.); famp@utad.pt (F.P.); jmorais@utad.pt (J.M.); 2Escola Superior de Tecnologia do Instituto Superior Politécnico de Viseu, Campus Politécnico de Repeses, 3504-510 Viseu, Portugal; 3UNIDEMI, Department of Mechanical and Industrial Engineering, NOVA School of Science and Technology, NOVA University Lisbon, 2829-516 Caparica, Portugal; 4Faculdade de Engenharia da Universidade do Porto, Departamento de Engenharia Mecânica e Gestão Industrial, Rua Roberto Frias, 4200-465 Porto, Portugal; mfmoura@fe.up.pt

**Keywords:** wood, cohesive law, digital image correlation, fracture mechanics, mixed mode I+II loading

## Abstract

Governing cohesive laws in mixed mode I+II loading of *Pinus pinaster* Ait. are directly identified by coupling the mixed mode bending test with full-field displacements measured at the crack tip by Digital Image Correlation (DIC). A sequence of mixed mode ratios is studied. The proposed data reduction relies on: (i) the compliance-based beam method for evaluating strain energy release rate; (ii) the local measurement of displacements to compute the crack tip opening displacement; and (iii) an uncoupled approach for the reconstruction of the cohesive laws and its mode I and mode II components. Quantitative parameters are extracted from the set of cohesive laws components in function of the global phase angle. Linear functions were adjusted to reflect the observed trends and the pure modes (I and II) fracture parameters were estimated by extrapolation. Results show that the obtained assessments agree with previous experimental measurements addressing pure modes (I and II) loadings on this wood species, which reveals the appropriateness of the proposed methodology to evaluate the cohesive law under mixed mode loading and its components.

## 1. Introduction

Structural applications of wood and wood products have been increasing recently, owing to economic and ecological reasons. As a result, the development of adequate failure criteria models becomes an imperative task. In this context, fracture mechanics-based approaches reveal to be most suitable, taking into consideration the anisotropic and heterogeneous nature of wood. Most of the works addressing this topic have been dedicated to fracture under pure mode loading, I or II [1,2,3,4,5,6]. In real structural applications, however, mixed mode conditions arise from external loading and/or because of the anisotropy and heterogeneity of the material. In this context, the mixed mode fracture characterization of wood becomes a relevant research topic, which has been addressed by several authors proposing different test methods. Jernkvist [7] employed the Double Cantilever Beam (DCB) with asymmetrical arms and the Single Edge Notched (SEN) tensile tests to investigate mixed mode I+II in Picea abies. The asymmetrical Wedge Splitting test was proposed by Tschegg et al. [8] to fracture the characterization of spruce under mixed mode I+II. In this test, the mode ratio can be altered using asymmetrical wedges with different angles. The referred tests only provide a very limited range of mode ratio values. In addition, fracture characterization was performed at initiation and results may suffer from some uncertainty on the definition of critical load.

A valuable alternative is the employment of the mixed mode bending (MMB) test, which consists of a combination of the DCB and End Notched-Flexure (ENF) tests through a specific conceived setup. The MMB was firstly proposed to characterize the fracture behavior in mixed mode I+II of polymeric composite materials [9,10]. The original set-up was then redesigned to mitigate erroneous toughness measurements introduced by geometric non-linearity during the test [11], and by the weight of some components [12]. Eventually, the MMB test turned out a standard for unidirectional composite materials [13]. The major advantage of the MMB test, over counterpart mixed mode tests, is allowing a spectrum of different mixed mode ratios by simply varying the length of the loading lever. The MMB test has already been applied to wood and wood bonded joints fracture characterization under mixed mode I+II loading, allowing the definition of their fracture envelopes [14,15].

Several full-field deformation techniques have been proposed and developed in recent decades for experimental solid mechanics [16]. A milestone was achieved by allowing contact free full-field kinematic measurements over a spectrum of length scales. The availability of this type of information has been shifting boundaries in several research domains. Different engineering problems have already been tackled such as experimental evidence of local gradients due to material heterogeneities [17,18], cracking characterization [19,20], damage and fracture model evaluation [21,22], high strain rate characterization of advanced composite material [23,24], validation of phenomenological numerical models [25] and multi-parameter identification from single test configurations [26,27]. Among the universe of full-field optical techniques, digital image correlation (DIC) was selected in this work [28]. DIC can be conveniently coupled with conventional apparatus such as testing machines, and typically only requires a careful preparation of a random pattern (with suitable speckled size and distribution) over the surface of interest. Moreover, the spatial resolution of the technique is flexibly adjusted by selecting the optical system (camera and lens), allowing to image a spectrum of different length scales of interest. Taking advantage of non-contact and full-field data, DIC can be suitable for fracture mechanics studies [29]. It can be used to access the local displacements field near the crack tip and to evaluate the crack tip opening displacement during the fracture tests [30]. This local information can therefore be used to determine the so-called cohesive law relating the tractions and relative displacements occurring at the crack tip. The cohesive law is representative of material fracture behavior, and is currently used in finite element analysis of materials fracture. The utilization of DIC to measure local displacement at the crack tip provides a direct approach, with the advantage of not imposing a priori the shape of the softening law.

The direct evaluation of the cohesive laws describing the mixed mode I+II fracture behavior of *Pinus pinaster* Ait. was experimentally assessed by coupling the MMB test to DIC measurements. Wooden specimens oriented in the RL propagation system were analyzed. The controlled parameters in the configuration of the MMB setup were set to address a range of mixed mode ratios. The compliance-based beam method (CBBM) was applied to independently determine the total strain energy release rate during the fracture tests, by only processing specimen dimensions, load and applied displacement. DIC measurements were post-processed to inspect the crack tip opening displacements at the initial crack location. Combining this information, the direct evaluation of the cohesive laws was assessed by a numerical approximation and differentiation approach. The evolution of the cohesive laws and its mode I and II components as function of the global phase angle was obtained, as well as the relations representing the evolution of the relevant cohesive parameters with the mode mixity.

## 2. Materials and Methods

### 2.1. Mixed mode Bending (MMB) Test

The MMB test was employed to study mixed mode I+II fracture loading. The geometry and nominal dimensions of the MMB specimen are shown in Figure 1a. The setup and free-body diagrams associated with the MMB test are presented in Figure 1b.

From the loading applied to the specimen, the energy of deformation due to bending can expressed as
(1)$$U=\int_{0}^{a}{\left(\frac{c-L}{2L}\right)}^{2}\frac{{\left(Px\right)}^{2}}{2E_{L}I}dx+\int_{0}^{a}{\left(\frac{c}{L}\right)}^{2}\frac{{\left(Px\right)}^{2}}{2E_{L}I}dx\;+\int_{a}^{L}{\left(\frac{c+L}{2L}\right)}^{2}\frac{{\left[P\left(x-a\right)\right]}^{2}}{16E_{L}I}dx+\int_{L}^{2L}{\left(-\frac{c+L}{2L}\right)}^{2}\frac{{\left[P\left(x-L\right)\right]}^{2}}{16E_{L}I}dx$$
where *P* is the applied load, $E_{L}$ is the longitudinal modulus, *I* is the second moment of area of each arm given by $I=Bh^{3}/12$, and *c* and *L* are geometric dimensions as defined in Figure 1. Algebraic manipulation yields the following expression:(2)$$U=\frac{P^{2}}{16E_{L}Bh^{3}L^{2}}\left[a^{3}\left(39c^{2}-18cL+7L^{2}\right)+2L^{3}{\left(c+L\right)}^{2}\right]$$

Applying the Castigliano Theorem, it is possible to obtain the displacement of the left extremity of the loading lever (Figure 1b), i.e., δ=dU/dP. The ratio between this displacement (δ) and applied load (P) is a measure of the specimen compliance (C):(3)$$C=\frac{1}{8E_{L}Bh^{3}L^{2}}\left[a^{3}\left(39c^{2}-18cL+7L^{2}\right)+2L^{3}{\left(c+L\right)}^{2}\right]$$

In order to avoid the difficult and inaccurate crack length monitoring during its propagation, an equivalent crack length-based procedure can be employed. With this aim, Equation (3) can be used to estimate the actual crack length (*a_e_*) as function of the current specimen compliance:(4)$$a_{e}={\left[\frac{8E_{L}Bh^{3}L^{2}C-2L^{3}{\left(c+L\right)}^{2}}{\left(39c^{2}-18cL+7L^{2}\right)}\right]}^{\frac{1}{3}}$$

The evolution of the total strain energy release rate ($G_{T}$) as function of the equivalent crack length can be obtained combining the Irwin-Kies relation
(5)$$G_{T}=\frac{P^{2}}{2B}\frac{dC}{da}$$
where *B* stands for the width of the beam (Figure 1a), with Equation (3), which leads to:(6)$$G_{T}=\frac{3P^{2}a_{e}^{2}}{16E_{L}B^{2}h^{3}L^{2}}\left(39c^{2}-18cL+7L^{2}\right)$$

This equation provides the evolution of the total strain energy release rate under mixed mode I+II loading as a function of $a_{e}$ for the MMB test, only measuring the applied load and displacement of the left extremity of the loading lever in the course of the test.

### 2.2. Evaluation of Cohesive Law

In the identification of a mixed mode cohesive law, it is typically assumed that total the strain energy release rate is a function of both components of the crack tip opening displacement thorough a given potential function Φ [31]:(7)$$G_{T}=\Phi \left(v,u\right)\;$$

Assuming an uncoupled approach, mode I (σ) and mode II (τ) stress components of the mixed mode cohesive law can be determined from the partial derivatives of the potential function as:(8)$$\sigma \left(v,u\right)=\frac{\partial \Phi }{\partial v}\;\;\;\;\;\;\;\;\;\;\;\;\;\mathrm{and}\;\;\tau \left(v,u\right)=\frac{\partial \Phi }{\partial u}$$

The phase angle (ϕ) is given by the ratio between the crack tip opening displacements (CTOD) components: tanφ=v/u. This term can be normalized, taking a global phase angle (θ) defined as
(9)$$\tan\theta =\frac{u_{c}}{v_{c}}\tan\varphi =\frac{u_{c}}{v_{c}}\cdot \frac{v}{u}$$
where $v_{c}$ and $u_{c}$ are the (average) critical or ultimate values of the CTOD in mode I and mode II, respectively. These reference values were determined independently from DCB [5] and ENF [6] tests carried out on the same wood species. The measurement of the crack tip opening displacement was achieved by post-processing displacements over a pair of subsets, selected regarding a coordinate system located at the initial crack tip with a spatial resolution of about 0.5 mm [5,6]. The assumption under the uncoupled modelling is supported by the scenario of a monotonically increment of the opening displacement at the crack tip [31]. Therefore, the components of the cohesive laws under mixed mode loading can be determined independently by using Equation (8) for a spectrum of phase angles.

The identification method proposed by Högberg [32] for the cohesive laws in mixed- mode I+II was used in this work. One assumption of this approach is the linearity of local deformation path during the mixed fracture test (i.e., constant mode ratio during the test). Hence, the evolution of the strain energy release rate will be only dependent on the magnitude of the total displacement (Δ), simple defined as:(10)$$\Delta^{2}=v^{2}+u^{2}$$

A second assumption on the theory is the existence of a potential function $G^{*}\;\left(\lambda \right)$ for each global phase angle θ (Equation (9)), from which the mixed mode cohesive law can be directly computed:(11)$$S_{\theta }\left(\lambda \right)=\frac{dG^{*}\left(\lambda \right)}{d\lambda }$$

In this equation $G^{*}$ represents the normalized total strain energy release rate defined as:(12)$$G^{*}\left(\lambda \right)=\frac{G_{T}\left(\lambda \right)}{2G_{c}}$$

The 2 factor is introduced by the linear assumption on the cohesive law and λ is the normalized crack tip displacement, which was assumed in the present work as:(13)$$\lambda =\frac{\Delta }{\Delta_{c}}$$

The explicit form of Equation (13), where the normalization is performed considering the specimen displacement measurements (Δ), instead of using reference critical values for mode I ($v_{c}$) and mode II ($u_{c}$) as initially proposed by Högberg [32], is justified by the dispersion typically found in biological materials such as wood. Finally, the individual mode I and mode II components of the mixed mode cohesive law can be decomposed according to the following relationships [32]
(14)$$\sigma \left(v\right)=\sigma_{u}\left(\frac{v}{v_{c}}\right)\frac{S}{\lambda }\;\;\;\;\;\;\;\;\;\;\;\mathrm{and}\;\;\;\;\;\;\;\;\;\;\tau \left(u\right)=\tau_{u}\left(\frac{u}{u_{c}}\right)\frac{S}{\lambda }$$
where $\sigma_{u}$ and $\tau_{u}$ are the cohesive strength values in mode I and mode II, respectively.
(15)$$G_{I}=\int_{0}^{v}\sigma \left(v,u\right)dv\;\;\;\;\;\;\;\;\;\;\;\;\;\;\mathrm{and}\;\;\;\;\;\;\;\;\;\;\;\;\;\;G_{\mathrm{II}}=\int_{0}^{u}\tau \left(v,u\right)du$$

## 3. Experimental Work

### 3.1. Material and Specimens

Logs from a single *Pinus pinaster* Ait. tree were selected for this work. In the first stage, boards were conventionally kiln-dried, and then left drying for about four weeks in the hygrothermal conditions of the laboratory environment (temperature between 20 and 25 °C and relative humidity between 60 and 65%). An average wood density of 643 kg/m^3^ was determined for an equilibrium moisture content of 12.3%.

Wooden specimens (Figure 2a) for the MMB set-up (Figure 2b) were manufactured with nominal dimensions of 2*h* = 20 mm, *L* = 230 mm, $L_{1}$ = 250 mm, $a_{0}$ = 162 mm and *B* = 20 mm, as shown in Figure 1a. Specimens were oriented to respect the RL propagation system. The initial crack on the specimens was introduced in two steps. Firstly, a notch of 1 mm thickness was sawn. Secondly, a final sharp crack with an extension of about 2–5 mm was introduced by manual impact using a thin blade.

### 3.2. Full-Field Deformation Measurements by DIC

The MMB set-up was coupled with digital image correlation. A suitable speckled pattern was initially painted over the wooden specimens (Figure 2b). A regular and thin layer of matt white was firstly painted over the natural surface of wood. A textured, random pattern was then created using an airbrush (IWATA, model CM-B, Anesta Iwata Iberica SL, Barcelona, Spain) with a 0.18-mm nozzle. The typical size of the speckles was imaged over at least three pixels to avoid image aliasing. The ARAMIS v6.0.2-6 DIC-2D system was used for image grabbing and image processing. A 8-bit Baumer charge coupled device (CCD) camera coupled with a TC2336 bi-telecentric lens was selected for the optical system. A region of interest of 29.3 × 22.1 mm^2^ was imaged, circumscribing the initial crack tip. The image focus was achieved by adjusting the working distance between specimen and lens to 103 mm. This image system defines a fixed conversion factor of 0.018 mm/pixel. Images were recorded during the test at a frequency of 1 Hz. The analogical load signal was synchronized with images during the tests. The DIC parameters were selected to enhance spatial resolution, since crack tip displacement measurements were required. A subset facet of 15 × 15 pixels^2^ and subset step of 13 × 13 pixels^2^ were selected. This set of parameters defined a displacement spatial resolution of 0.270 mm. The resolution or accuracy of the DIC measurements was estimated as the standard deviation of the Gaussian noise signal, which is typically measured on pair of images recorded on an in-plane rigid-body translation of the speckled pattern [33]. A sub-pixel displacement resolution in the order of 10^−2^ pixel was obtained.

### 3.3. MMB Fracture Tests

The MMB set-up used in this work is described in [14] (see Figure 2b). This setup has the advantage of allowing the variation of mixed mode I+II ratio by adjusting the distance *c* in the setup (Figure 1b). In this study, several mixed mode ratios were selected. The MMB fracture tests were carried out in a universal testing machine (model 1125, Instron, Barcelona, Spain) at a controlled displacement rate of 0.5 mm/min. The applied load was measured by means of a 100 kN load cell. The signals of the load were simultaneously recorded by a HBM SPIDER 8 with a frequency of 10 Hz.

## 4. Results and Discussion

From the raw load-displacement (P–δ) curves, the resistance-curves (*R*–curves) were determined by means of the CBBM approach described in Section 2.1. As discussed in Section 2.2, the methodology adopted to evaluate the cohesive laws assumes uncoupled behavior between mode I and mode II.

For the purpose of the direct evaluation of the cohesive laws, both P–δ and P–CTOD curves were obtained from raw data by coupling the MMB test with DIC measurements. The CTOD was naturally decomposed into mode I (v) and mode II (u) components to define P–v and P–u curves, respectively. For this evaluation, a pair of points were just selected above and below the initial crack tip location to evaluate the relative displacement during test [5,6]. Figure 3 shows the average evolution of CTOD as a function of δ, for each interval of global phase angles analyzed in this work. As expected, both u and v components of the CTOD are strictly increasing functions of δ. These examples are representative of the expected relative amplitudes of u and v regarding the mixed mode ratios of the MMB specimens. It was noticed that the amplitude of v compared to u increases when the phase angle increases revealing a predominance of mode I loading.

The local deformation path at the crack tip was expressed by the u–v relationship, as summarized in Figure 4 for the given phase angles ranges. As it can be seen, the monotonic variation was typically linear, which reinforces the statement of an almost constant mode-mixity during loading. Consequently, the global phase angle (Equation (9)) for each mixed mode ratio was determined by linear regression over the data points until the maximum load, as shown in Figure 4.

Figure 5 shows a typical example of the $G^{*}\left(\lambda \right)$ function (Equation (12)). To compute the derivative of this function with regard to λ (Equation (11)), three types of continuous functions were firstly fitted to the raw data using the least-square regression method. The purpose of this fitting is both filtering intrinsic experimental noise and providing a robust mathematical framework for the differentiation. Since the $G^{*}\left(\lambda \right)$ curve has a typical *S*-shape (sigmoid curve), theoretically converging to a plateau which represents the critical strain energy release rate, the logistic function was initially selected
(16)$$G^{*}=\frac{A_{1}-A_{2}}{1+{\left(\lambda /\lambda_{0}\right)}^{p}}+A_{2}$$
where $A_{1}$, $A_{2}$, p and $\lambda_{0}$ are constants determined by the least-square regression method. Secondly, a cubic smoothing spline estimate ${\hat{G}}^{*}\left(\lambda \right)$ of the function $G^{*}\left(\lambda \right)$ was selected, as defined to be the minimizer of the objective function [34]
(17)$$p\underset{i}{\sum }w_{i}{\left[G_{i}^{*}-{\hat{G}}^{*}\left(\lambda_{i}\right)\right]}^{2}+\left(1-p\right)\int^{\;}{\left(\frac{d^{2}{\hat{G}}^{*}}{d\lambda^{2}}\right)}^{2}d\lambda$$
for specified smoothing parameter p (0≤p≤1) and weights ($w_{i}$). For small values of p, the fitted curve tends to be too rigid, and for values close to unity (with a significant number of digits), the regression curve loses the filtering capacity. In the analysis, a smoothing parameter p = 0.989 was chosen, along with weights $w_{i}=1$ for all data points.

Lastly, the Prony-series was also considered, expressed by the following relationship [34]
(18)$$\frac{G^{*}}{G_{\max}^{*}}=\overset{n\sum^{\;}\left(-\frac{n\lambda }{i\lambda_{\max}}\right)}{\underset{i=1}{\sum }}B_{i}exp$$
where the parameters $B_{i}$ in the series are determined by least-square regression. In Equation (18), $G_{max}^{*}$ represents the maximum value of $G^{*}$, and $\lambda_{max}$ is the corresponding value of λ. In a sensitivity study, the terms of n were taken between 7 and 17. A convergence was achieved by considering n=10.

An application example of the typical curve fitting analysis by least-square regression and reconstructed curves (by differentiation) are shown in Figure 5. The resulting cohesive laws components for mode I and mode II are shown, respectively, in Figure 5c,d. It can be observed that the Prony-series tend to underestimate the initial and final parts of the raw data. On the other hand, the flexibility of the smoothing spline to follow the raw data comes at the cost of a reduced filtering effect. Finally, the logistic function guarantees a certain degree of smoothing, and moreover, forces the regularization of the typical $G^{*}\left(\lambda \right)$
*S*-shape curve. Furthermore, the differentiation can be computed directly from the fitted constants defining the function (Equation (15)). Consequently, the logistic curve was considered for the systematic analysis of the data.

The mode I and mode II components of the reconstructed cohesive laws were determined according to Equation (14). Figure 6 shows an overview of mean curves for each phase angle intervals considered. Overall, it can be stated that the mode I component of the cohesive law increases with the increase of the global phase angle (Figure 6a), in contrast to what happens with the mode II component (Figure 6b). This behavior agrees with the anticipated trend, which reinforces the appropriateness of the proposed methodology.

In more detail, Figure 7 plots the evolution of the several cohesive parameters as function of the global phase angle. The rising trend of the normal peak stresses (Figure 7a) and the decreasing tendency observed for the shear ones were approximated by linear functions. Extrapolating for θ=90° in the linear fitted equation (σ=f(θ)) Figure 7a and for θ=0° in the relation of Figure 7b provides an estimation of the local strengths under pure mode I and pure mode II, respectively. The obtained values point to $\sigma_{u}$ = 8.96 MPa and $\tau_{u}$ = 16.56 MPa. These values compare well with the ones reported in [4] for the same wood species ($\sigma_{u}$ = 7.93 MPa and $\tau_{u}\;$= 16.0 MPa), revealing that the followed procedure effectively captures these material parameters.

The same strategy was followed for the evolution of the strain energy release rate components ($G_{I}$ and $G_{\mathrm{II}}$) in function of θ. Considering the linear relationships obtained in Figure 7c,d, the critical fracture energies point to $G_{IC}\;$= 0.5 N/mm and $G_{IIC}\;$= 1.0 N/mm, respectively. In the mode I case, the obtained value is somewhat higher than expected, since a recent characterization of this wood species under mode I loading [35] pointed to values in the range of 0.3–0.4 N/mm. In contrast, the $G_{IIC}$ value is in close agreement with recent pure mode II fracture characterization, which points to $G_{IIC}\;$= 0.97 N/mm [36].

## 5. Conclusions

This work addresses the experimental identification of cohesive laws in mixed mode I+II of *Pinus pinaster* Ait. The approach combines the MMB test with DIC measurements. The MMB setup was configured to target different mixed mode ratios. The obtained results showed that the ratio of the local mixed mode (tanϕ=v/u), determined by opening displacements in mode I (v) and mode II (u), is almost constant throughout the test. This observation allowed the definition of the global phase angle for each mixed mode ratio, assuming linear regressions. It was found that mode I components of the cohesive laws under mixed mode reveal an increase, with the global phase angle in opposition to what happens with the mode II component. The evolution of the peak stresses and strain energy release rate components as functions of the global phase angle was analyzed in more detail. It was observed that normal peak stress and mode I strain energy release rate component increase with the global phase angle, contrasting to shear peak stress and mode II strain energy release rate component that decrease alongside it. These relations were fitted by linear functions that were subsequently used to estimate the pure mode I and II parameters by extrapolation. The values obtained for the local strengths ($\sigma_{u}$ and $\tau_{u}$) and critical fracture energies ($G_{\mathrm{IC}}$ and $G_{\mathrm{IIC}}$) are globally in agreement with previous experimental determinations for this wood species.

These results validate the proposed procedure as a valuable tool to assess the cohesive laws and its components, and serve as a support for development of cohesive zone models appropriate to deal with the mixed mode fracture behavior of wood and other materials.

## Figures and Tables

**Figure 1 materials-14-00374-f001:**
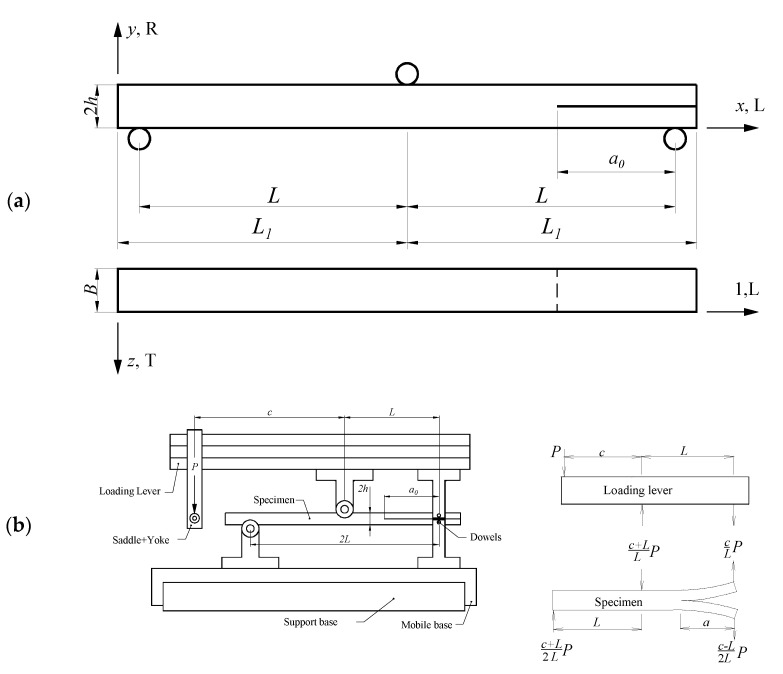
Mixed mode bending (MMB) test: (**a**) specimen geometry (2*h* = 20 mm, *L* = 230 mm, $L_{1}$ = 250 mm, $a_{0}$ = 162 mm and *B* = 20 mm); (**b**) set-up and free-body diagrams (*P*—applied load).

**Figure 2 materials-14-00374-f002:**
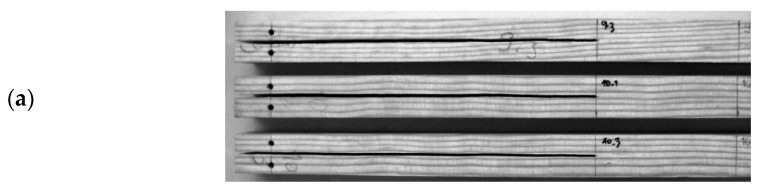
(**a**) MMB specimens oriented in the RL propagation system; (**b**) MMB set-up coupled with digital image correlation (DIC) (speckle pattern and its histogram over a region of interest of 29 × 20 mm^2^).

**Figure 3 materials-14-00374-f003:**
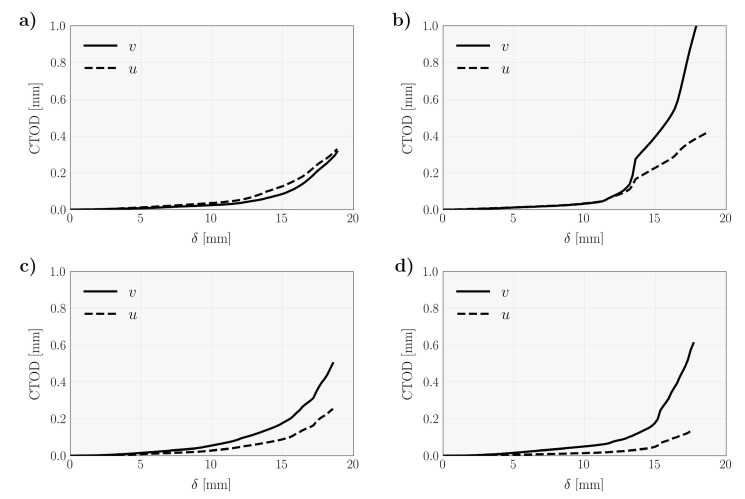
Average crack tip opening displacement (CTOD) as a function of the applied displacement (v: mode I component of the CTOD; u: mode II component of CTOD), for the global phase angles (**a**) $45^{{}^{\circ}}<\theta <60^{{}^{\circ}}$; (**b**) $60^{{}^{\circ}}<\theta <70^{{}^{\circ}}$; (**c**) $70^{{}^{\circ}}<\theta <80^{{}^{\circ}}$; and (**d**) $80^{{}^{\circ}}<\theta <90^{{}^{\circ}}$.

**Figure 4 materials-14-00374-f004:**
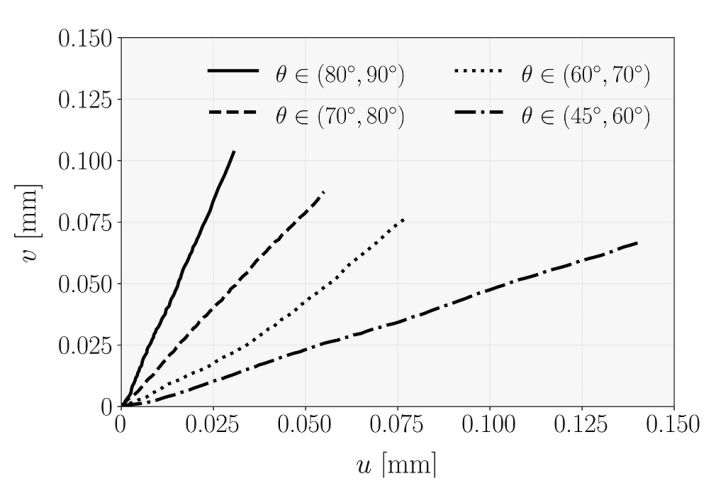
Pattern of the local deformation for each mixed mode ratio of the MMB tests.

**Figure 5 materials-14-00374-f005:**
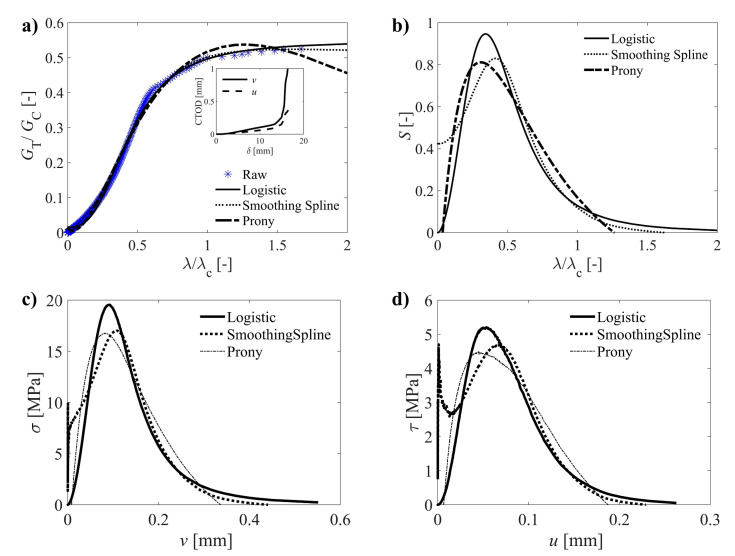
For a typical specimen: (**a**) potential function; (**b**) normalized cohesive law; (**c**) mode I; and (**d**) mode II components of the reconstructed cohesive laws.

**Figure 6 materials-14-00374-f006:**
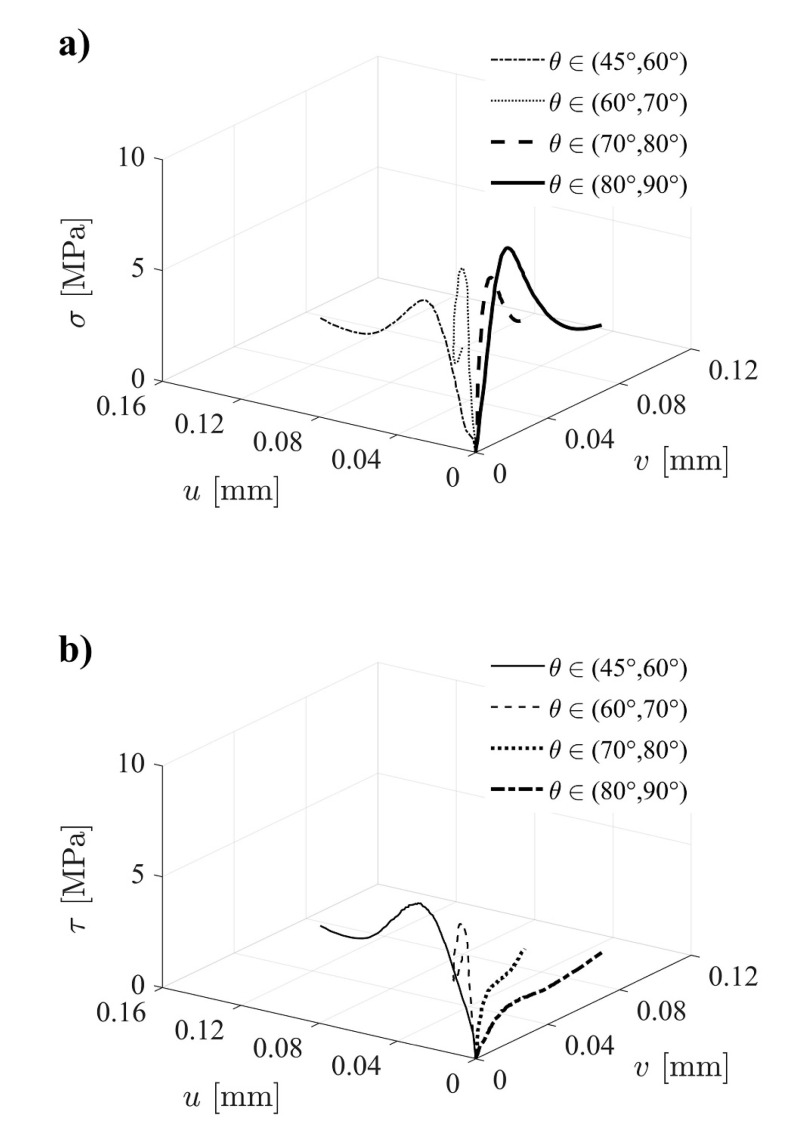
Mean cohesive laws distribution over the range of phase angles: (**a**) σ(v,u);  (**b**) τ(v,u).

**Figure 7 materials-14-00374-f007:**
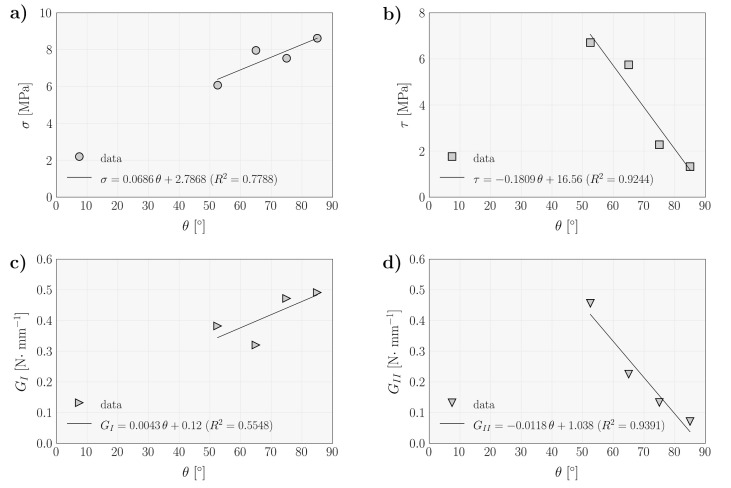
Evolution of cohesive law parameters as a function of global phase angle (**a**) ultimate normal stress; (**b**) ultimate shear stress; (**c**) mode I strain energy component; and (**d**) mode II strain energy component.

## Data Availability

Data sharing not applicable.

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
