# Peer review of "Direct Evaluation of Mixed Mode I+II Cohesive Laws of Wood by Coupling MMB Test with DIC"

_materials, 2021, doi:10.3390/ma14020374_

Round 1

Reviewer 1 Report

A fes speel checks to be done: a index o and mm2 (exponent) mm².

Fig. a lower part - fig. is mirro inverted

Introduction. Write ot ENF test, when used the first time.

Fig. 1 and and caption text (line 95): L1 -  L1 (1 as subscript) is not consistent in text and Figure, check subscript of ao (o as subscript) in text and figure. "P .. load" should be written in the caption text.
Note: P for load should be changed to F (force), as P is the symbol for Power (W).

Line 179 .. mm² instead of mm2, or 29 mm x 20 mm

b or B for specimen widths?. Dimensions should be used consistently: L is the symbol of the dimension, but l, h, b are the physical variable.

Is the cracklength somewhere defined?

Author Response

Manuscript ID: materials-1048310

Title: Direct evaluation of mixed mode I+II cohesive laws of wood by coupling MMB test with DIC

Journal: Materials (ISSN 1996-1944)

Section: Structure Analysis and Characterization

The authors would like to thank the reviewers for the valuable comments provided to the original version of the manuscript. Accordingly, all questions have been addressed with the view to improve the manuscript. A revised version of the paper is prepared, in which carefully considerations of all queries mentioned in the reviewers' comments were included. Changes are highlighted in blue colour in the new version of the paper. Please, find enclosed a point-by-point response to the reviewer’s comments.

Reviewer 1

Comments and Suggestions for Authors

A fes speel checks to be done: a index o and mm2 (exponent) mm².

Fig. a lower part - fig. is mirro inverted

Authors: These formatting corrections are amended in the revised manuscript. Fig 1b was unexpectedly reversed when printing to the pdf file.

Introduction. Write ot ENF test, when used the first time.

Authors: These corrections are amended in the revised manuscript.

Fig. 1 and and caption text (line 95): L1 - L1 (1 as subscript) is not consistent in text and Figure, check subscript of ao (o as subscript) in text and figure. "P .. load" should be written in the caption text.

Note: P for load should be changed to F (force), as P is the symbol for Power (W).

Authors: The correct formatting of “L1” and “a0“are amended in the revised manuscript. The authors have always used the symbol “P” for the load value since it is typically used by the scientific community in the framework of fracture mechanics. Therefore, for the sake of consistency the symbol “P” has kept in the manuscript, but it is clearly reinforced, both in the text and Fig. 1 content, that "P” refers to “.. load".

Line 179 .. mm² instead of mm2, or 29 mm x 20 mm

Authors: The correct formatting is amended in the revised manuscript.

b or B for specimen widths?. Dimensions should be used consistently: L is the symbol of the dimension, but l, h, b are the physical variable.

Authors: A new sentence is introduced in the revised manuscript to clarify the meaning of B in Eq. (5), as the width of the beam; the height of the beam is referred to 2h (Figure 1a).

Is the cracklength somewhere defined?.

Authors: The initial crack length is defined in section “3.1. Material and specimens”: = 162 mm. The CBBM data reduction does not require the measurement of the crack length during the test. Therefore, no more considerations were addressed in the paper. However, the DIC method could be used to measure the crack length variation during the propagation fracture test (see Ref [5,19,20]).

Reviewer 2 Report

Dear authors,

your paper contains novel, interesting and significant information, however it should be investigated and revised according following suggestions;

  1. The definition of some parameters are unclear or properly stated. Please investigate again carefully;
    Parameters in equation (1), P,
    u,v (original point and direction),
    C in eq. (5) should be small ?
    what is phai in eq. (9)

  2. what is "critical" in equation (9)
  3. Fig 1. (b) lower right should be mirrored.
  4. Insert measure in Fig 2. (b) right
  5. The Plot Curves (lines) of Fig 4 are derived from the curves in Figure 3., by certain points (critical ?) .

  6. The explanation for c) and d) in Fig 5. is missing or incomplete.
  7. Is there any limitation for u and v ? If the deflection of the specimen is too large, u and v deviate from each coordinate direction. Or are they defined by local coordinate system ?
  8. Could you please explain more the relationships between Figure 6 and equation (8) ?

Author Response

Manuscript ID: materials-1048310

Title: Direct evaluation of mixed mode I+II cohesive laws of wood by coupling MMB test with DIC

Journal: Materials (ISSN 1996-1944)

Section: Structure Analysis and Characterization

The authors would like to thank the reviewers for the valuable comments provided to the original version of the manuscript. Accordingly, all questions have been addressed with the view to improve the manuscript. A revised version of the paper is prepared, in which carefully considerations of all queries mentioned in the reviewers' comments were included. Changes are highlighted in blue colour in the new version of the paper. Please, find enclosed a point-by-point response to the reviewer’s comments.

Reviewer 2

Dear authors,

your paper contains novel, interesting and significant information, however it should be investigated and revised according following suggestions;

The definition of some parameters are unclear or properly stated. Please investigate again carefully;

Parameters in equation (1), P,

u,v (original point and direction),

C in eq. (5) should be small ?

what is phai in eq. (9)

what is "critical" in equation (9)

Fig 1. (b) lower right should be mirrored.

Authors: The correct formatting is amended in the revised manuscript, considering Eqs. (1), (5) and (9). The parameter C in Eq. (5) stands for Compliance and should not be confused with c, which is a geometric dimension (Fig 1.c). All parameters in Eq. (9) are defined in the text, including \phi and the critical values for v  and u are referring to ultimate values of the CTOD. Fig 1.b was unexpectedly inverted in the printed pdf file.

Insert measure in Fig 2. (b) right

Authors: In the caption of Fig. 2(b) right, regarding the region of interest captured by the digital camera for the DIC measurements, dimensions are defined in the text has highlighted in the revised manuscript. Otherwise, the figure just shows the histogram of the speckled pattern as a qualitative measure of its quality, in which all relevant dimensions or measure are given (in pixel units).

The Plot Curves (lines) of Fig 4 are derived from the curves in Figure 3., by certain points (critical ?) .

Authors: Plots in Fig. 4 are averaged values over the defined intervals for the angle q. A sentence was added in the revised manuscript to clarify this point.

The explanation for c) and d) in Fig 5. is missing or incomplete.

Authors: A new sentence is added in the revised manuscript stating that: “The resulting cohesive laws components for mode I and mode II are shown, respectively, in Figures 5 c) and d).”

Is there any limitation for u and v ? If the deflection of the specimen is too large, u and v deviate from each coordinate direction. Or are they defined by local coordinate system ?

Authors: The measurement of the u and v displacements are measured locally regarding the initial crack tip location. To clarify this point, a sentence was rewritten in the revised manuscript as: “The measurement of the crack tip opening displacement was achieved by post-processing displacements over pair of subsets, selected regarding a coordinate system located at the initial crack tip with a spatial resolution of about 0.5 mm”.

Could you please explain more the relationships between Figure 6 and equation (8) ?

Authors: Eq. (7) represents a mathematical surface from which the partial derivatives with respect to both v and u, expressed in Eq. (8), directly yield mode I (σ) and mode II (τ) stress components of the mixed-mode cohesive law. Nevertheless, the curves in Fig. 6 are determined from Eq. (14), which has been obtained following the proposed data reduction presented in the Section “2.1. Mixed-mode bending (MMB) test”.

Reviewer 3 Report

See attached file.

Author Response

Manuscript ID: materials-1048310

Title: Direct evaluation of mixed mode I+II cohesive laws of wood by coupling MMB test with DIC

Journal: Materials (ISSN 1996-1944)

Section: Structure Analysis and Characterization

The authors would like to thank the reviewers for the valuable comments provided to the original version of the manuscript. Accordingly, all questions have been addressed with the view to improve the manuscript. A revised version of the paper is prepared, in which carefully considerations of all queries mentioned in the reviewers' comments were included. Changes are highlighted in blue colour in the new version of the paper. Please, find enclosed a point-by-point response to the reviewer’s comments.

Reviewer 3

The study deals with the cohesive laws in mixed I+II mode bending loading of solid wood (Pinus pinaster). The study, using direct DIC determination, researched a sequence of mixed mode rations. Quantitative parameters were extracted from the set of cohesive laws. Trends were confirmed in the pure I and II fracture parameters. The manuscript confirmed the appropriateness of the experimental and theoretical principles to evaluate cohesive law under mixed mode loading.

The manuscript has proper “IMRaD” structure. Introduction properly presents past studies. Methodology, theoretical and experimental, is correct, and well explained. Conclusive remarks are well supported by the research results.

Remarks and comments:

L161 “In the first stage, boards were artificially dried in an oven,…” It is better to use “conventionally kiln dried”, if the wood was dried technically, using convective kiln dryer.

Authors: This sentence is improved accordingly in the revised manuscript.

L169 “Firstly, a notch of 1 mm thickness was swan,” The sentence is not understandable. Please specify correctly, how the notch was made? Do you mean sawn?

Authors: This sentence is corrected in the revised manuscript.

L243 “Secondly, a cubic smoothing spline estimate G*(λ) of the function G*(λ) was selected,…” It is not understandable at the moment – are the symbols correctly written?

Authors: This sentence is corrected accordingly in the revised manuscript.